# Learning to Generate Formally Verifiable Step-by-Step Logic Reasoning via Structured Formal Intermediaries

## Abstract

Large language models (LLMs) have recently demonstrated impressive performance on complex, multi-step reasoning tasks, especially when post-trained with outcome-rewarded reinforcement learning (Guo et al., 2025). However, it has been observed that outcome rewards often overlook flawed intermediate steps, leading to unreliable reasoning steps even when final answers are correct. To address this unreliable reasoning, we propose *PRoSFI* (*Process Reward over Structured Formal Intermediates*), a novel reward method that enhances reasoning reliability without compromising accuracy. Instead of generating formal proofs directly, which a modest-size (7B) model can rarely do successfully, the model outputs structured intermediate steps aligned with its natural language reasoning. Each step is then verified by a formal prover. Only fully validated reasoning chains receive high rewards. The integration of formal verification guides the model towards generating step-by-step machine-checkable proofs and hence yields more credible final answers. *PRoSFI* offers a simple and effective approach to training trustworthy reasoning models.

## 1 Introduction

Recent advances in techniques such as Chain-of-Thought (CoT) prompting and Reinforcement Learning (RL) have significantly enhanced the reasoning capabilities of large language models (LLMs). These methods have improved both the length and quality of model-generated reasoning chains, giving rise to the paradigm of test-time scaling (Snell et al., 2024). Representative systems include OpenAI o1 (Jaech et al., 2024) and DeepSeek-R1 (Guo et al., 2025), which set new records across multiple reasoning benchmarks.

Notably, DeepSeek-R1-Zero (Guo et al., 2025), which employs a rule-based reward mechanism (outcome correctness plus output format score) to guide learning via the Group Relative Policy Optimization (GRPO) algorithm, successfully achieves strong multi-step reasoning capabilities and exhibits emergent reflection behaviors. The simplicity and efficiency of this approach have inspired a series of follow-up works, such as R1-Zero replications TinyZero (Pan et al., 2025) and Logical RL (Xie et al., 2025), and various GRPO variants (Yu et al., 2025; Liu et al., 2025).

However, although these approaches greatly improve the final accuracy, they do not guarantee the correctness of the reasoning steps. In mathematical tasks, it has been observed that models may arrive at correct answers through invalid arguments (Uesato et al., 2022; Lightman et al., 2024). Empirical studies, including our own observations, show that RL guided by outcome-only rewards can degrade the faithfulness of reasoning chains (Chen et al., 2025b). Furthermore, in domains like proof generation, where final results cannot be automatically validated, outcome-based rewards are inapplicable. To ensure trustworthy reasoning, process-level supervision, not just outcome supervision, is necessary.

Process-level supervision provides feedback at each chain-of-thought step, yielding more precise training signals and producing models with more interpretable, reliable reasoning steps (Uesato et al., 2022). Nonetheless, collecting large-scale process-level feedback is challenging as human annotation at the granularity of individual proof steps is prohibitively expensive (Lightman et al., 2024). A natural solution is to leverage formal provers to verify intermediate reasoning steps. This

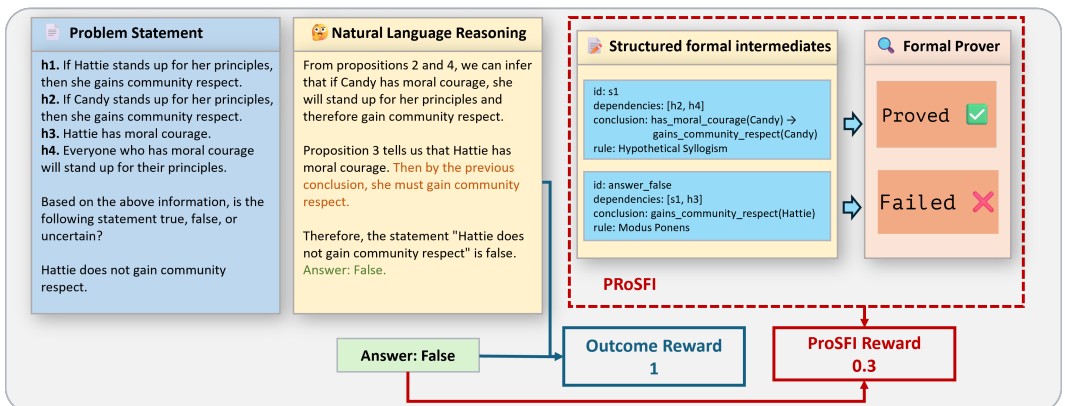

Figure 1: **Pipeline for *Process Reward over Structured Formal Intermediates* and comparison with Outcome Reward.** The process begins with a problem statement (left block) containing multiple propositions (h1–h4) related to individuals (Hattie and Candy). Then, a large language model (LLM) first generates natural language reasoning steps to solve the problem (middle block). The **Outcome Reward** approach then directly compares the generated answer with the ground truth, assigning a reward of 1 if they match. In contrast, *PRoSFI* introduces an additional step by generating structured formal intermediates that capture logical dependencies, inferred conclusions, and the application of formal inference rules such as *Hypothetical Syllogism* and *Modus Ponens*. These formal representations are then submitted to a **Formal Prover**, which verifies the logical soundness of the reasoning. In the illustrated example, although the generated answer ("Hattie does not gain community respect" is false) is correct, the formal proof fails verification. As a result, *PRoSFI* assigns a low reward despite the correct outcome.

approach requires access to formal codes, either directly generated by language models or translated from their natural language reasoning steps, which poses two key challenges.

First, models that excel in natural language reasoning often struggle to generate correct formal code. Training models to write formal proofs typically requires much larger model sizes and significant data distillation (Wang et al., 2025; Ren et al., 2025). Moreover, models optimized for formal proof generation often struggle with structure and format following (Xin et al., 2024). Consequently, it is difficult to have a model capable of generating both natural language explanations and aligned formal proofs simultaneously.

Second, existing models that excel at formal proof generation typically are not designed to take natural language inputs, and there is a lack of reliable methods for translating natural language into formal code, as well as for evaluating the faithfulness of such translations. Furthermore, generating formal code is inherently more challenging for large language models than producing natural language reasoning. Accurate formal code generation often relies on test-time scaling techniques like Best-of-N (BoN), which can significantly increase computational costs particularly during RL post-training.

To address these challenges, we propose a novel and effective reward method: *Process Reward over Structured Formal Intermediates* (*PRoSFI*, see Figure 1). Instead of generating full formal proofs directly, *PRoSFI* asks the model to output a series of reasoning blocks where each block contains natural language explanation and a corresponding structured intermediate representations. This intermediate representation takes the form of a formatted array, usually in JSON or YAML, including an identifier, a logical rule, a derived conclusion (in formal language), and a list of dependencies (prior steps or premises). Then, formal propositions are extracted from each intermediate representation and passed to a formal prover for verification. The reasoning chain is considered trustworthy and receives the highest reward only if all steps are successfully verified.

Specifically, *PRoSFI* offers the following advantages. First, generating intermediate representations is easy. Existing models, even modest-size (7B) language model, are typically well-equipped for generating structured intermediate representations, and in this case, they only need to produce formalized propositions for each intermediate step—without having to construct full formal proofs.

This avoids the core challenges of formal proof construction. Second, structured formal intermediates facilitate formal verification. Specifically, formal provers are tasked with verifying individual propositions derived from the intermediate representation, rather than constructing complete proofs for the original problems. This ensures step-level alignment with the intermediate reasoning. Overall, the structured formal intermediates serve as an effective bridge between language models and formal provers. They reduce the burden on both sides: language models are freed from producing formal syntax, avoiding distribution drift, while provers benefit from simplified sub-problems defined in a detailed and verifiable format.

To evaluate the effectiveness of our method, we conduct controlled experiments in a logical reasoning setting. We use the ProverQA dataset (Qi et al., 2025b), which is hard for current models and, as a synthetic dataset, provides high-quality formal representations of the problems. Through extensive experiments, we demonstrate that *PRoSFI* can significantly improve the reasoning soundness for complex logical problems. We summarize our contributions as follows:

- **Conceptually,** we introduce a new paradigm: reinforcement learning with step-by-step rewards guided by formal-provers, which, to our knowledge, has not been addressed in literature. This shifts the focus from rewarding only final answers to ensuring the correctness of the entire reasoning process. We create a setup for training and evaluating verifiable reasoning models of a modest size (7B), greatly lowering the entry barrier for the community.

- **Technically,** we propose *PRoSFI*, a framework that effectively bridges language models and formal provers. It enables RL post-training by incorporating formal verification feedback into policy optimization.

- **Empirically,** we demonstrate that Qwen2.5-7B-Instruct, after *PRoSFI* RL post-training, can generate verifiable proofs on hard first-order logic tasks, achieving this without relying on massive LLMs or extensive data distillation, marking the first such result at this scale.

## 2 RELATED WORK

**Reinforcement Learning Post-Training** has recently played a crucial role in training large language models, particularly for human preference alignment (RLHF) and enhancing complex reasoning abilities. Early work such as InstructGPT (Ouyang et al., 2022) leveraged Proximal Policy Optimization (PPO) (Schulman et al., 2017) to extract reward signals from human preference data, steering the model toward outputs more aligned with user expectations. However, the high computational cost of PPO on large-scale models motivated the development of more efficient alternatives.

One such alternative is Group Relative Policy Optimization (GRPO) (Shao et al., 2024), which has been adopted in training DeepSeek-R1-Zero (Guo et al., 2025). DeepSeek-R1-Zero trains a pretrained model by using GRPO with only rule-based rewards to achieve significant gains on complex mathematical reasoning tasks. Since then, many replication projects become public Pan et al. (2025).

**Logical Reasoning Datasets** serve as important benchmarks for evaluating the deductive and symbolic reasoning capabilities of LLMs. These datasets are typically constructed either by manual annotation or via automatic generation. Manually curated collections, such as Reclor (Yu et al., 2020) and FOLIO (Han et al., 2022), offer rich linguistic diversity and complex logical rules.Automatic dataset generation methods have been proposed. ProntoQA (Saparov & He, 2022) and ProofWriter (Tafjord et al., 2020) use predefined templates or rule-based systems to generate large-scale reasoning problems with controllable complexity.

Other synthetic logic-reasoning datasets also include LogicPro (Jiang et al., 2024), which converts LeetCode program problems into narrative reasoning problems and leverages Python execution to extract and verify detailed intermediate solution traces; LogicBench (Parmar et al., 2024b), a natural language inference benchmark that covers propositional, first-order, and non-monotonic logics; and FLD (Morishita et al., 2023; 2024b), which employs a set of first-order logic rules to generate synthetic multistep proofs.

**Formal-Language-Augmented Reasoning Techniques** have recently attracted great attention to improve language models' rigor and reliability. In logical reasoning tasks, systems such as LINC (Olausson et al., 2023) and Logic-LM (Pan et al., 2023) treat the LLM as a semantic parser, translating natural language premises and conclusions into first-order logic expressions. These expressions

are then checked by external solvers like Prover9 (McCune, 2005) or Z3 (De Moura & Bjørner, 2008). Logic-LM further incorporates a self-correction loop, using solver error feedback to iteratively refine the symbolic translations until they pass validation. Similarly, Logic Agent (Liu et al., 2024) systematically applies propositional logic rules to structure and verify the model's reasoning, enhancing both effectiveness and interpretability. A partial list of other related logical and program aid reasoning related work includes (Gao et al., 2023; Lightman et al., 2024; Morishita et al., 2024a; Olausson et al., 2023; Pan et al., 2023; Parmar et al., 2024a; Qi et al., 2025a; Ryu et al., 2025; Xu et al., 2024; Yang et al., 2023; Zhou et al., 2024a).

For automated theorem proving, OpenAI's GPT-f system (Polu & Sutskever, 2020) first demonstrated that LLMs can generate formal mathematical proofs, achieving a 31% success rate on the miniF2F benchmark (Zheng et al., 2022). More recent advances along this direction includes DeepSeek-Prover-V1.5 (Xin et al., 2024), Kimina-Prover (Wang et al., 2025), DeepSeek-Prover-V2 (Ren et al., 2025), and Seed-Prover (Chen et al., 2025a). Delta-Prover (Zhou et al., 2025) further leverages the sketching and reflection capabilities of general-purpose LLMs, solving 95.9% of the MiniF2F problems through decomposition and iterative reflection.

In mathematical reasoning, recent work has begun to integrate provers into LLM reasoning: Don't Trust: Verify (DTV) framework (Zhou et al., 2024b) focuses on quantitative solutions by instructing LLMs to formalize their natural language steps into Isabelle (Paulson, 1994) proof scripts, which are then automatically verified. On benchmarks such as GSM8K (Cobbe et al., 2021) and MATH (Hendrycks et al., 2021), DTV outperforms naive majority voting by filtering out unsound solution paths, demonstrating that formal provers can supply valuable feedback for reasoning paths.

**Process Reward Models and Verification.** Early work (Uesato et al., 2022; Lightman et al., 2024; Wang et al., 2023) train process reward models (PRMs) to provide fine-grained feedback for reasoning steps, achieving higher accuracy on GSM8K and MATH benchmarks. Recent research has shifted towards using LLMs as verifiers to detect errors in the reasoning process. For example, Huang & Yang (2025) employed a general-purpose model with a verification-refinement pipeline to achieve a gold medal in IMO 2025.

## 3 Preliminaries on Dataset, Base Model, Algorithms and Evaluation Metrics

**Dataset Description** ProverQA (Qi et al., 2025b) is a synthetic first-order logic reasoning benchmark generated by the ProverGen framework, which combines the linguistic diversity of LLMs with the rigor of a symbolic prover (Prover9 (McCune, 2005)). More details are deferred to Appendix B.

**Model Description** The Qwen2.5-7B-Instruct model, an instruction-tuned variant of the Qwen2.5 family developed by Alibaba's Qwen team, comprises 7 billion parameters and is released under the Apache License 2.0 (Team, 2024). It builds on the Qwen2.5 base LLM with further fine-tuning to follow instructions and engage in chat/dialogue. Specifically, it can produces reliable structured outputs (e.g. JSON), parses tables, spreadsheets, and other semi-structured formats robustly.

**Group Relative Policy Optimization (GRPO)** GRPO is a variant of Proximal Policy Optimization (PPO) tailored for training large language models (LLMs) without an explicit value (critic) model (Shao et al., 2024). The algorithm details can be found in Appendix C.

**Metrics** We introduce several metrics to evaluate the various facets of the model performance: *Answer Correct Rate*, *Reward Hit Rate* and *Soundness*.

| Metric | Answer Correct Rate | Reward Hit Rate | Soundness |
|---|---|---|---|
| **Definition** | $\dfrac{\#\text{Final Answer Correct}}{\#\text{Total Examples}}$ | $\dfrac{\#\text{Outputs with Max Reward}}{\#\text{Total Examples}}$ | $\dfrac{\#\text{Answer Correct \& Path Valid}}{\#\text{Total Examples}}$ |

Table 1: Evaluation metrics and their definitions.

*Answer Correct Rate* measures the percentage of examples where the model produces the correct final answer, regardless of the validity of the reasoning steps. *Reward Hit Rate* reflects how often the model receives the maximum reward ($R = 1$) under our reinforcement learning framework, that is, when the final answer is correct and all intermediate reasoning steps are formally verified. *Soundness* assesses both answer correctness and reasoning path validity, which is defined as the proportion of examples where the final answer is correct and the reasoning path is logically valid.

We note that *Reward Hit*, i.e., all structured formal intermediates are verified, does not necessarily guarantee the *Soundness* of the entire reasoning path. This is because the extracted formal propositions may not fully cover the entire reasoning path. In practice, rigorously assessing the soundness of a reasoning path would require extensive human evaluation, which is often infeasible at scale. As an alternative, we adopt state-of-the-art LLM-based evaluators, such as GPT-OSS-120B, to approximate *Soundness*. In the experimental section, we will analyze potential biases introduced by these LLM-as-a-Judge metrics.

With the dataset, model, algorithm, and evaluation metrics defined, we now explore how to train LLMs to generate step-by-step reasoning that is formally verifiable.

## 4 METHOD: *Process Reward over Structured Formal Intermediates*

To incorporate step-level verification into the reinforcement learning process, a natural approach is to (1) ask for the model emit natural language reasoning step and formal proof script at the same time (e.g., in Lean 4 (Moura & Ullrich, 2021)), and (2) use an theorem prover to verify the correctness of each step. The verification outcomes can then be used as reward signals to distinguish valid reasoning chains from unsound ones.

**However, direct generation of stepwise formal proofs fails in practice.** We prompted the model to emit Lean 4 proof code immediately after natural-language reasoning steps, exploring multiple prompt templates and both Qwen2.5-7B-Instruct and Qwen2.5-Coder-7B-Instruct variants. However, these attempts yielded unsatisfactory results: only a small fraction of the generated Lean scripts compiled successfully, and those that did were overly reliant on generic tactics such as *simp*. The proofs were typically brittle, poorly structured, and misaligned with the model's natural-language reasoning, and consequently ineffective for verifying the reasoning process.

We attribute this failure to two key challenges. First, modest-sized models (e.g., 7B) struggle to generate accurate and meaningful formal proofs. This makes it difficult for GRPO to bootstrap useful supervision from the generated formal codes, a finding consistent with recent progress in formal proof generation, which has primarily relied on much larger base models (Wang et al., 2025; Ren et al., 2025).

Second, the process of integrating formal verification into the GRPO rollout pipeline introduces significant computational overhead. The frequent calls to external formal provers disrupt GPU parallelism during sequence generation, drastically slowing down training and reducing scalability.

These limitations motivate our alternative solution: rather than having models directly generate full formal proofs, we introduce a structured intermediate representation that enables effective extraction of formal propositions and step-level verification with formal tools, which at the same time remaining within the capability of modest-sized language models.

### 4.1 THE DESIGN OF *Process Reward over Structured Formal Intermediates*

To overcome the challenges encountered in directly generating formal proofs, we propose a novel approach: *Process Reward over Structured Formal Intermediates* (*PRoSFI*), which introduces a structured JSON/YAML based intermediate representation as a bridge between natural language reasoning paths and formal verification. This format serves as a lightweight, verifiable scaffold that enables formal reward during reinforcement learning. Crucially, Qwen2.5-7B-Instruct, the base model, is well-suited for this task due to its strong formatting capabilities, particularly for structured JSON or YAML outputs (Team, 2024).

**Structured Formal Intermediates**   In *PRoSFI*, the model is prompted to produce a sequence of reasoning steps in structured format, where each step explicitly encodes the structure of logical reasoning, which includes the following fields: `id`: a unique identifier for the step, used for reference; `dependencies`: a list of premises this step depends on, which may include given conditions or previous steps; `conclusion`: The new conclusion drawn in this step, expressed in a formal language; `rule`: The logical rule applied to derive the conclusion.

To ensure each step remains atomic, we impose a constraint that the number of dependencies per step must be fewer than 5 (tunable). Each step can then be converted into a sub-problem for verification. Moreover, we also require the final step's conclusion must match the target proposition, and its identifier must be among the provided answer candidates.

This structured format allows natural language reasoning to be decomposed into discrete, formalizable sub-problems. For the first-order logic (FOL) tasks considered in this paper, which are relatively simple and well-structured, each proposition can typically be verified using lightweight tactics in Lean (e.g., `aesop`, `simp`), or through external automated solvers such as Z3 (De Moura & Bjørner, 2008) or Prover9 (McCune, 2005). For more complex setting, we optionally delegate verification to powerful formal models such as DeepSeek-Prover-V1.5 (Xin et al., 2024). These configurations can be adapted with different formal formats or verification tools for different tasks and domains.

**Reward Design**   If all steps pass formal verification, the entire reasoning path will be assigned with a high reward $R = 1$. Otherwise, a failure of any individual step flags a potential flaw in the reasoning process. Based on this framework, we define the reward function as follows:

$$R = \begin{cases} 1.0, & \text{Answer correct and all steps verified,} \\ 0.3, & \text{Answer correct, but some steps not verified,} \\ 0.1, & \text{Output format correct, but answer incorrect,} \\ 0.0, & \text{Output format incorrect or other failure cases.} \end{cases}$$

This reward design of *PRoSFI* exploits the rigor of formal prover while avoiding the difficulties of generating full formal proofs, resulting in more stable, fine-grained, and structure-aligned supervision signals, which are crucial for effective reinforcement learning. Moreover, *PRoSFI* avoids the efficiency bottlenecks typically associated with integrating formal provers into the rollout phase of RL training. The model generates the full natural language reasoning and structured formal intermediates in a single, uninterrupted forward pass. Formal verification is performed after generation, allowing it to be fully decoupled from the autoregressive process. This decoupling not only improves training throughput by enabling batch-level parallelism but also allows verification tools to run asynchronously, maximizing efficiency.

We next present empirical results to demonstrate that *PRoSFI* can effectively improve the soundness of LLM reasoning paths.

## 5 EXPERIMENTAL RESULTS

### 5.1 TRAINING AND TESTING SETUP

**Training Setup**   As described in Section 3, our experiments are conducted on the ProverQA dataset, a synthetic logical reasoning benchmark designed with controlled difficulty levels and well-defined ground-truth labels (True/False/Uncertain). This makes it particularly well-suited for evaluating models in a controlled reinforcement learning (RL) setting. To address the difficulty imbalance within the ProverQA dataset, we excluded instances labeled as Uncertain. The remaining data was then partitioned into training and test sets for each difficulty level (easy, medium, and hard), yielding a total of 767 instances for the training set.

We adopt Qwen2.5-7B-Instruct as our base model. This model offers a best choice for performance given the model size constraint by hardware barrier. Moreover it has demonstrated superior responsiveness to RL-based post-training compared to other open-source alternatives such as the LLaMA series Shao et al. (2025); Zeng et al. (2025).

We use the GRPO algorithm with a learning rate of $3 \times 10^{-7}$ and batch size of 32. For each training iteration, GRPO samples 16 responses, with the maximum sequence length set to 5120 tokens. The KL coefficient in the GRPO loss function is set to 0.001. To enhance training stability, we implement two strategies from the Decoupled Advantage Policy Optimization (DAPO) algorithm (Yu et al., 2025), whose details are in Appendix C.

All experiments are conducted on a compute node equipped with 8× Hopper GPUs, running on Debian GNU/Linux 12 with CUDA 12.4, PyTorch 2.6.0, vLLM 0.8.3, and Verl 0.4.0. Each training run consists of 960 iterations, which takes approximately 14 hours to complete.

**Testing Setup**   Besides the test splits of difficulty levels in the ProverQA dataset, we introduce a more challenging out-of-distribution (OOD) test set, named ProverQA-Extra. This set consists of 158 synthesized examples generated from the ProverQA repository using Qwen2.5-32B-Instruct as the background story generation model. To reduce randomness, we sampled 16 responses for each test data instance and took the average of their results.

## 5.2   OUTCOME REWARD LEADS TO HIGH ANSWER CORRECT RATE BUT LOW SOUNDNESS

We employ GRPO with a reward scheme similar to that of DeepSeek-R1-Zero to replicate its training methodology on ProverQA. This serves a baseline for the evaluation of reasoning path soundness.

**Reward Design**   Following DeepSeek-R1-Zero pipeline, the model must emit its chain of thought wrapped in `<think>` ... `</think>` tags before the final answer, referred to as *format correct*. We define the outcome-based reward $R$ as:

$$R = \begin{cases} 1.0, & \text{format correct and answer correct,} \\ 0.1, & \text{format correct but answer incorrect,} \\ 0.0, & \text{format incorrect.} \end{cases}$$

This reward encourages adherence to the output format and correctness of the final answer.

**Prompt Design**   The system prompt, adapted from DeepSeek-R1-Zero, is provided in Appendix D.

The user prompt starts with the original dataset description, then presents the problem in formal language with clearly indexed premises (see Figure 1). We refer to this setup as *Outcome Reward with CoT* or *Outcome-CoT* for short. The full user prompt is presented in Appendix D.

**Results: GRPO with *Outcome-CoT* significantly boosts the Answer Correct Rate but achieves rather low Soundness.**   Specifically, in our evaluation on the ProverQA-Hard subset, GRPO with *Outcome-CoT* substantially improves the Answer Correct Rate, rising from 46.0 with Qwen2.5-7B-Instruct to 94.89. This confirms the successful replication of the R1-Zero training procedure. However, the GPT Soundness score remains low, improving only from 9.0 to 21.31. This indicates that although the final answers are often correct, the generated reasoning paths frequently contain flawed or invalid inference steps.

## 5.3   DIRECT LEAN GENERATION FAILS TO PROVIDE RELIABLE STEPWISE VERIFICATION

As a direct baseline for integrating formal verification into reinforcement learning, we evaluated an approach in which the model is required to emit complete Lean 4 proof scripts (rather than the structured intermediates introduced in *PRoSFI*). This baseline was designed to isolate the effect of output modality: reward schedule and training protocol were kept the same as in the structured-intermediate condition, and the only difference was that the model produced full formal proofs immediately following each natural-language reasoning trace.

Empirically, this direct-generation strategy failed to provide a useful stepwise verification signal for GRPO. Compile-and-verify rates were extremely low: the Reward Hit Rate on ProverQA-Hard was 0.00% and on ProverQA-Extra was 2.69%. Instances that did compile were vanishingly rare, and when compilation succeeded the resulting proofs typically reduced to a single opaque tactic (e.g. a one-line `simp` invocation) that bore little structural correspondence to the model's natural-language chain of thought. Consequently, the direct-Lean baseline produced almost no reliable, fine-grained

supervision: both final-answer accuracy and measured reasoning soundness decreased relative to the Outcome-CoT baseline, as shown in table 2).

## 5.4 *PRoSFI* LEARNING WITH PROVERQA

**Results: Structured formal rewards improve final accuracy with significantly more credible reasoning path.** After GRPO training with stepwise formal rewards, we achieve comparable final-answer accuracy to the GRPO training with only final-answer reward on ProverQA, as shown in Table 2. This confirms that our stepwise rewards can effectively be combined into R1-Zero reward with high reasoning accuracy. We can also see that stepwise formal rewards can significantly improve the credibility of reasoning paths in comparison with final-answer only reward in Section 5.2. Moreover, the results on ProverQA-Extra also demonstrates *PRoSFI* achieves strong out-of-distribution (OOD) performance on logical reasoning questions of previously unseen difficulty.

| Method | Metric (%) | Hard | Extra |
|--------|-----------|------|-------|
| Outcome-CoT | Answer Correct Rate | 94.89 | 92.52 |
| | GPT Soundness | 21.31 | 8.15 |
| Direct Lean Generation | Answer Correct Rate | 92.71 | 91.93 |
| | GPT Soundness | 20.45 ↓ −0.9 | 6.29 ↓ −1.9 |
| ProSFI | Answer Correct Rate | 92.99 | 85.36 |
| | GPT Soundness | 74.05 ↑ 52.7 | 28.72 ↑ 20.6 |

Table 2: Results on ProverQA Hard and Extra[1]. GPT Soundness values include the difference of each method versus the Outcome-CoT baseline.

## 5.5 TEST-TIME SCALING BEHAVIORS OF *PRoSFI*

A critical advantage of *PRoSFI* is its strong *test-time scaling* behavior, made possible through integration with the *Don't Trust; Verify* (DTV) method (Zhou et al., 2024b). DTV is designed to enhance both accuracy and reasoning reliability by sampling multiple reasoning paths and employing a lightweight verifier to assess the logical validity of each output. If any candidates are successfully verified, one of them is selected; otherwise, the method defaults to majority voting. *PRoSFI* is particularly well-suited for integration with DTV due to its structured intermediate representations, which enable reliable formal verification. In contrast, approaches that rely solely on natural language reasoning lack the structured outputs required for effective collaboration with DTV, limiting their ability to benefit from test-time verification.

In our experiments, we apply the DTV method to the outputs generated by *PRoSFI*. As shown in Figure 2, increasing the number of sampled reasoning paths consistently improves performance on the Soundness metric, demonstrating *PRoSFI*'s robust and scalable behavior at test time. This trend holds across both the `ProverQA-Hard` and `ProverQA-Extra` subsets. In contrast, methods such as `Outcome-CoT` fail to benefit from additional sampling, as majority voting alone is insufficient to discern the soundness of the reasoning paths and cannot effectively filter out logically invalid reasoning.

---

[1]After analyzing the failure cases, we found that ProverQA contains non-trivial data quality issues (e.g., identical natural-language propositions being formalized with different predicates; see AppendixJ for details). On a manually repaired subset of the extra split, ProSFI's Answer Correct Rate / GPT Soundness increase from 85.36% / 28.72% to 90.80% / 40.10%, while Outcome-CoT reaches 93.00% / 11.40% and is less affected because it relies more on natural language than on formal proofs. A fully fair comparison would require retraining all methods on a fully repaired ProverQA, which we are currently working on.

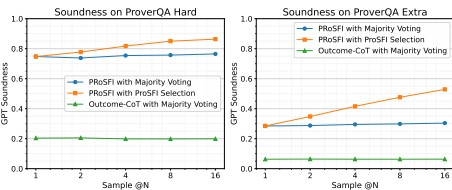

Figure 2: **Left:** GPT Soundness on `ProverQA-hard` subset. **Right:** GPT Soundness on `ProverQA-extra` subset. As the number of sampled paths increases, *PRoSFI* with *PRoSFI* selection consistently improves, while `Outcome-CoT` cannot benefit due to translation limits.

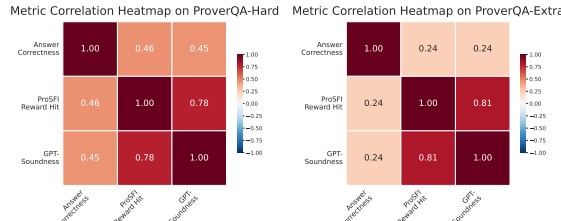

Figure 3: **Left:** Metric correlation heatmap on `ProverQA-Hard`. **Right:** Metric correlation heatmap on `ProverQA-Extra`. Compared to Answer Correctness, *PRoSFI* Reward Hit better reflects reasoning-path soundness.

### 5.6 *PRoSFI* REWARD ALIGNS WELL WITH SOUNDNESS

To evaluate the effectiveness of the reward design used in *PRoSFI*, we conduct a correlation analysis among three metrics: Answer Correctness, *PRoSFI* Reward Hit, and GPT-Soundness of generated reasoning paths. This analysis is performed on the outputs of *PRoSFI*, and the results are in Figure 3.

The results reveal that the *PRoSFI* Reward Hit signal exhibits a high correlation with GPT-Soundness, much higher than the Answer Correctness. This suggests that our reward function is more aligned with step-by-step reasoning quality, well achieving the design goal. In contrast, answer correctness alone may fail to capture subtle flaws or inconsistencies in intermediate reasoning steps. These findings highlight that *PRoSFI*'s reward design provides meaningful supervision that better promotes logically coherent and sound outputs.

## 6 *PRoSFI* GENERALIZES TO KNIGHTS AND KNAVES DATA

To demonstrate the generalization ability of our proposed method, we extend experiments to the Knights and Knaves dataset Xie et al. (2024), whose details can be found in Appendix B.3.

For the Outcome-CoT baseline, we adopt the same prompt template as Logic-RL, while the prompt for *PRoSFI* is adapted slightly to suit this task. Following the out-of-distribution (OOD) evaluation protocol established in Logic-RL, our model is trained exclusively on instances with difficulty levels from 3ppl to 7ppl. Evaluation is then conducted across the full test range, from 2ppl to 8ppl, with the 8ppl split serving as the primary benchmark for OOD performance.

The training configuration closely follows that of ProverQA, including a batch size of 32, 16 sampled responses per prompt, a maximum sequence length of 5120, and the same KL coefficient and optimization strategies from DAPO, except that we set the learning rate to $4 \times 10^{-7}$, consistent with Logic-RL, and extend the training schedule to 1920 steps to align with our experimental design.

| Method | Metric (%) | 2ppl | 3ppl | 4ppl | 5ppl | 6ppl | 7ppl | 8ppl |
|---|---|---|---|---|---|---|---|---|
| Outcome-CoT | Answer Correct Rate | 99.94 | 99.94 | 99.94 | 99.50 | 98.38 | 92.56 | 89.75 |
| | GPT Soundness | 76.81 | 79.06 | 71.13 | 70.31 | 63.75 | 58.69 | 50.88 |
| ProSFI | Answer Correct Rate | 99.25 | 100.00 | 99.31 | 99.63 | 96.94 | 91.88 | 85.31 |
| | GPT Soundness | 87.44 | 83.81 | 79.44 | 76.63 | 77.50 | 64.63 | 58.75 |
| | | ↑ 10.6 | ↑ 4.8 | ↑ 8.3 | ↑ 6.3 | ↑ 13.8 | ↑ 5.9 | ↑ 7.9 |

Table 3: Results on Knights and Knaves dataset.

From the results shown in Table 3, we can see that on the Knights and Knaves dataset, ProSFI consistently improves GPT soundness over Outcome-CoT by 5–14 points across all difficulty levels, while maintaining comparable answer accuracy, demonstrating strong logical consistency and reli-

able generalization, particularly on the hardest 8ppl split. Therefore, our approach is generalizable to other logical reasoning tasks beyond ProverQA.

## 7 CONCLUSION

It is known that LLMs may make mistakes in the reasoning steps even when the final answer is correct, which undermine its credibility in high-risk domains. In this work, we study the problem of how to enable large language models generating step-by-step verifiable reasoning paths via formal prover's feedback. We propose a novel method *Process Reward over Structured Formal Intermediates* that connects natural language reasoning with formal provers' verification. By training LLM to output structured formal intermediates and validating them with formal provers, we obtain precise, step-level soundness signals. These signals serve as stepwise formal rewards during reinforcement learning, enabling the model to learn verifiably correct reasoning behaviors.

While our current framework focuses on first-order logic tasks, future work can extend it to broader domains such as general natural language reasoning and automatic theorem proving. These extensions will require more diverse datasets, meticulously designed cold-start strategies, stronger base models, and deeper integration with domain-specific formal reasoning tools.

Another viable direction involves further exploration of training algorithm development. Instead of relying solely on binary pass/fail signals, future systems could incorporate more granular reward functions or more precise credit assignment mechanisms to enhance training efficacy and stability.

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
