# Appendices

## A  USE OF LARGE LANGUAGE MODELS (LLMS)

While preparing this manuscript, we made use of Large Language Models (LLMs) as writing aids, specifically for tasks such as grammar correction, wording refinement, and rephrasing to enhance clarity and readability. In line with ethical guidelines, we, the authors, confirm that we take full responsibility for the content of this work. Any text produced with the assistance of LLMs has been carefully reviewed, revised, and approved by us. All scientific claims, findings, and conclusions are entirely our own. We are responsible for any possible mistakes, inaccuracies, or ethical issues that may arise in this submission.

## B  DETAILS OF DATASETS

### B.1  PROVERQA DATASET

The ProverQA dataset contains 1,500 problems evenly split into easy, medium, and hard subsets, each defined by controlled complexity. For each difficulty level, we performed a custom train-test split: 400 examples for training and 100 for testing. To assess generalization to even harder problems, we further synthesized 100 additional extra hard examples using the same framework, referred to as the *Extra Split*.

Every instance of ProverQA comprises:

- A background story created for each problem to introduce linguistic context and variation,

- A logic skeleton of FOL premises (facts + rules with 7 operators: $\wedge, \vee, \neg, \rightarrow, \oplus, \forall, \exists$) and distractors (non-essential facts or semantically related but irrelevant rules),

- A goal to classify as *True, False, or Uncertain*,

- Intermediate reasoning chains provided in both symbolic form and natural language.

This design meets key criteria: scalability, natural & diverse language, explicit symbolic representations, and faithful reasoning chains. At the same time, it is challenging enough, for example, GPT4o achieves accuracy 46.2 with standard prompts, and 50 with CoT prompts for hard samples (Qi et al., 2025b). Hence it is good to work with ProverQA for logical reasoning studies.

### B.2  AN EXAMPLE FROM PROVERQA

To illustrate the structure of the ProverQA dataset, we present a representative example as follows. Each entry is organized as a JSON object containing the question, multiple-choice options, the correct answer, and the reasoning chain used to derive the conclusion. In addition, the dataset provides a context field capturing the relevant facts and rules, as well as a mapping from natural language (NL) to first-order logic (FOL) formulas (nl2fol). Finally, the conclusion_fol field specifies the logical statement under evaluation.

This example highlights how ProverQA combines natural language reasoning with its corresponding logical formalization, enabling the evaluation of models not only on final predictions but also on their formal reasoning steps.

**An Example from ProverQA**

```
{
  "id": 0,
  "options": [
    "A) True",
    "B) False",
    "C) Uncertain"
  ],
  "answer": "B",
  "question": "Based on the above information, is the following statement true, false, or
    ↪ uncertain? Brecken has never experienced heartbreak.",
  "reasoning": "fact1: Brecken has experienced heartbreak.\nrule: Either Brecken has
    ↪ experienced heartbreak or he has never experienced heartbreak, but not both.\
    ↪ nconclusion: Brecken has experienced heartbreak.\n\nTherefore, it is false that
    ↪ Brecken has never experienced heartbreak. The correct option is: B.",
  "context": "Brecken has experienced heartbreak. Either Brecken has experienced
    ↪ heartbreak or he has never experienced heartbreak, but not both.",
  "nl2fol": {
    "Brecken has experienced heartbreak.": "has_experienced_heartbreak(Brecken)",
    "Either Brecken has experienced heartbreak or he has never experienced heartbreak, but
    ↪   not both.": "has_experienced_heartbreak(Brecken) ⊕
    ↪ has_never_experienced_heartbreak(Brecken)"
  },
  "conclusion_fol": "has_never_experienced_heartbreak(Brecken)"
}
```

### B.3 KNIGHT AND KNAVES DATASET

The Knights and Knaves (K&K) puzzles(Xie et al., 2025) are an algorithmically generated reasoning dataset where characters are either knights, who always tell the truth, or knaves, who always lie. The goal is to determine each character's role from their statements. The dataset is procedurally generated, providing unlimited variation and unseen test cases for evaluating generalization. Difficulty is precisely controlled by varying the number of characters (2–8) and the complexity of logical operations, supporting curriculum learning and out-of-distribution testing. Each puzzle has a single verifiable solution, ensuring evaluations rely on strict deductive reasoning.

K&K dataset is particularly useful due to its formal, synthetic design. Every puzzle is governed by explicit rules and admits a unique, deterministic solution, eliminating ambiguity and allowing researchers to clearly separate true reasoning ability from memorization.

Here is an example adapted from the Xie et al. (2025).

**An Example from K&K Puzzle Problem**

```
Problem: A very special island is inhabited only by knights and knaves. Knights always
    ↪ tell the truth, and knaves always lie. You meet 2 inhabitants: Zoey, and Oliver.
    ↪ Zoey remarked, "Oliver is not a knight". Oliver stated, "Oliver is a knight if and
    ↪  only if Zoey is a knave". So who is a knight and who is a knave?

Solution: (1) Zoey is a knave (2) Oliver is a knight
```

## C ALGORITHM DETAILS

For completeness, we revisit the GRPO algorithm in detail. The procedure consists of the following steps:

1. Samples a group of output sequences $\{o_1, \dots, o_G\}$ for each prompt $q$ from the policy $\pi_{\theta_{\text{old}}}$.

2. Scores each sequence $o_i$ with a reward model $r_\phi(q, o_i)$, yielding rewards $\{r_i\}_{i=1}^G$.

3. Computes relative advantages by normalizing these rewards within the group:

$$\hat{A}_i = \frac{r_i - \frac{1}{G}\sum_{j=1}^G r_j}{\sqrt{\frac{1}{G}\sum_{j=1}^G \left(r_j - \frac{1}{G}\sum_{k=1}^G r_k\right)^2}}.$$

4. Optimizes the policy by maximizing a PPO-style clipped surrogate objective, with a KL penalty against a frozen reference model $\pi_{\text{ref}}$.

Moreover in the experiments, we implement two strategies from the Decoupled Advantage Policy Optimization (DAPO) algorithm (Yu et al., 2025) to stablize the training of GRPO. Specifically,

- **Clip Higher**: This strategy increases the upper clip range of the importance sampling ratio in the policy gradient loss to encourage exploration. Consistent with the DAPO methodology, we set the asymmetric clipping range with $\varepsilon_{\text{low}} = 0.2$ and $\varepsilon_{\text{high}} = 0.28$.

- **Overlong Penalty**: To reduce reward noise from excessively long outputs, a penalty is applied. Specifically, for responses where the length $l$ exceeds 4096 tokens, a linear penalty is imposed, calculated as $r_{\text{penalty}} = -\frac{l-4096}{5120-4096}$.

## D  PROMPTS FOR THE OUTCOME-COT APPROACH

---

**System Prompt for Outcome-CoT**

```
System:

Given a problem statement as contexts, the task is to answer a logical reasoning question.

The assistant first thinks about the reasoning process in the mind and then provides the
    ↪ user with the answer.
The reasoning process is enclosed within <think> </think> tags, followed by the answer, i.
    ↪ e., <think> reasoning process here </think> answer here.

The last A), B), or C) in the answer section (i.e., the content outside <think> </think>)
will be considered your answer. If the format is incorrect, you may not receive a score.
```

---

The User Prompt for the Outcome-CoT approach used in Section 5.2 is as follows.

The prompt begins with a context section containing background knowledge expressed in natural language, including facts (e.g., Paola appreciates beauty) and rules (e.g., Anyone who can solve problems and communicate effectively has practical skills). These statements establish the logical environment within which reasoning must take place.

Following the context, the prompt specifies a question that requires truth evaluation of a target statement—in this case, whether Paola is artistically inclined. The prompt also provides a set of options (True, False, or Uncertain), mirroring the multiple-choice format used throughout the dataset. Finally, the template ends with the phrase "The correct option is:", leaving space for the model to generate its answer.

864
865
866
867
868
869
870
871
872
873
874
875
876
877
878
879
880
881
882
883
884

**User Prompt for Outcome-CoT**

```
User:

Context:
Anyone who can solve problems and communicate effectively has practical skills.
If Paola appreciates beauty, then she values precision and enjoys manual work.
Paola appreciates beauty.
Paola is either skilled in mathematics or has practical skills, but not both.
If someone enjoys manual work and values attention to detail, then they can craft with
    ↪ their hands.
Paola is either skilled in mathematics or artistically inclined, but surprisingly, she is
    ↪ actually both.
Paola values attention to detail.
Anyone who crafts with their hands has practical skills.

Question:
Based on the above information, is the following statement true, false, or uncertain?
Paola is artistically inclined.

Options:
A) True
B) False
C) Uncertain

The correct option is:
```

885
886
887
888

# E  PROMPTS AND EXAMPLE RESPONSES FOR *PRoSFI*

889
890

## E.1  SYSTEM PROMPT FOR *PRoSFI*

891
892
893
894

To demonstrate how *PRoSFI* is guided in the reasoning process, we provide the system prompt used in Section 4 (see Listing X). The prompt instructs the model to behave as a mathematical reasoner that operates on premises expressed in Lean4 syntax (e.g., $\land, \lor, \neg, \rightarrow, \oplus, \forall, \exists$).

895
896

The structure of the prompt has several key elements:

897
898

- **Formatted reasoning**: All intermediate thinking is enclosed within <think>...</think>.

899
900
901

- **Formatted summarized output**: Inside <summary>...</summary>, the model must return a JSON array of atomic inference steps.

902
903

Each **JSON object** specifies:

904
905

- id: a unique identifier for the step,

906
907
908

- dependencies: the premises or earlier steps it relies on,

909
910

- conclusion: a single new FOL formula expressed in Lean4,

911
912

- rule: the inference rule applied.

913
914
915
916

**Constraints** – Each inference step must be as fine-grained as possible (fewer than 5 dependencies), and the JSON must be valid and parsable. The final object corresponds to the model's chosen answer and must align with one of the provided options.

917

This design enforces faithful, step-by-step reasoning, making the model's decision process transparent and verifiable.

**System Prompt for PRoSFI**

```
System:

You are a math reasoner. For each problem, you will receive a set of premises annotated in
    ↪  Lean4 (use →, ⊕, ∧, ∨, ¬, ∀, ∃). Answer the multiple-choice reasoning question
    ↪  as follows:

1. Internal reasoning goes inside <think>···</think>.
2. Then, inside <summary>···</summary>, emit a JSON array of atomic inferences, one per
    ↪  object:

  ```json
  {
    "id": "s",
    "dependencies": ["h₁","s"], // original premises or earlier steps
    "conclusion": "...",          // exactly one new FOL formula (Lean4 syntax)
    "rule": "..."                 // the single inference rule applied
  }
  ```

  - The JSON array must be valid and parsable without errors.
  - Each object must represent exactly one atomic inference step.
  - Each step should be as atomic as possible, using less than 5 dependencies.

The last object in the array corresponds to your answer. Its id and conclusion must be
    ↪  identical to one of the options.
```

## E.2 USER PROMPT FOR PRoSFI

To make the interaction format concrete, we show the **user prompt** used for *PRoSFI* in Section 4. The prompt is divided into several components:

- **Context**: A sequence of natural language premises paired with their corresponding formal statements in Lean4 syntax. For example, the first two premises assert conditional rules about Hattie and Candy, while the 3rd and 4th connect moral courage to principled action.

- **Question**: A query formulated in natural language, asking whether a target statement (here, "Hattie does not gain community respect") is true or false given the context.

- **Options**: The possible answers, each expressed both in natural language and as a formal Lean4 statement.

- **Answer slot**: The prompt ends with "The correct option is:", leaving space for the model to provide its choice.

This structure illustrates how *PRoSFI* is guided to align natural language reasoning with formal logical inference, ensuring that both the problem and the candidate answers are clearly specified in parallel in both natural language and formal statement.

**User Prompt for PRoSFI**

```
User:

Context:
1. If Hattie stands up for her principles, then she gains community respect. Formal
   ↪ statement: 'h₁ : stands_up_for_principles Hattie → gains_community_respect Hattie
   ↪ '.
2. If Candy stands up for her principles, then she gains community respect. Formal
   ↪ statement: 'h₂ : stands_up_for_principles Candy → gains_community_respect Candy'.
3. Hattie has moral courage. Formal statement: 'h₃ : has_moral_courage Hattie'.
4. Everyone who has moral courage will stand up for their principles. Formal statement: '
   ↪ h₄ : ∀x, (has_moral_courage x → stands_up_for_principles x)'.

Question: Based on the above information, is the following statement true or false? Hattie
   ↪  does not gain community respect.

Options:
A) True. Formal statement: 'h_goal_true: ¬gains_community_respect Hattie'.
B) False. Formal statement: 'h_goal_false: ¬(¬gains_community_respect Hattie)'.

The correct option is:
```

### E.3 Model Response after RL with PRoSFI

We next illustrate how the model responds after RL with PRoSFI. The response is divided into two main parts:

- **Reasoning path**: Enclosed within <think> ... </think>, the model provides a step-by-step explanation in natural language. In this case, it applies Universal Instantiation and Modus Ponens to derive that Hattie stands up for her principles, and subsequently that she gains community respect. This reasoning shows why the negated claim (Hattie does not gain community respect) is false.

- **Structured summary**: Enclosed within <summary> ... </summary>, the model outputs a JSON array of atomic inference steps. Each object specifies the inference id, the set of dependencies, the derived conclusion (in Lean4 syntax), and the rule applied. The last object corresponds to the final answer and matches one of the candidate options.

We can see the model's response is both interpretable in natural language and machine-verifiable through its structured reasoning steps.

```
Assistant:

<think>
1. From h₄ (∀x, has_moral_courage x → stands_up_for_principles x) and h₃ (has_moral_courage
   ↪ Hattie), by Universal Instantiation on Hattie and Modus Ponens, derive
   ↪ stands_up_for_principles Hattie.
2. From h₁ (stands_up_for_principles Hattie → gains_community_respect Hattie) and the above,
   ↪ by Modus Ponens, derive gains_community_respect Hattie.
3. Therefore ¬ gains_community_respect Hattie is false.
</think>
<summary>[
  {
    "id": "s1",
    "dependencies": ["h₃", "h₄"],
    "conclusion": "stands_up_for_principles Hattie",
    "rule": "Universal Instantiation + Modus Ponens"
  },
  {
    "id": "h_goal_false",
    "dependencies": ["h₁", "s1"],
    "conclusion": "¬(¬gains_community_respect Hattie)",
    "rule": "Modus Ponens"
  }
]
</summary>
```

## F   Typical Errors for Outcome-CoT

In this section, we select two representative cases to analyze two typical types of errors that occur during the reasoning process of the Outcome-CoT model, namely semantic confusion and commonsense shortcuts.

### F.1   Semantic Confusion

```
User:

Context:
...
3. Either Miracle is community-oriented or self-centered, but not both.
...
6. If Miracle is a leader, then she is either selfless or thinks of herself, but not both.
...
Assistant:
...
Since she is community-oriented, she cannot be self-centered (statement 3). Therefore, she
    ↪ must be selfless. Thus, the statement 'Miracle is not self-centered' is true.
...
```

In this example, the model infers that "Miracle is not self-centered" from the premise that "Miracle is community-oriented", and then equates "not self-centered" with "selfless". In doing so, it bypasses a rigorous logical analysis of the premises and directly concludes that "Miracle is selfless". However, determining whether Miracle is truly selfless requires a more comprehensive evaluation based on Proposition 6, whether Miracle is a leader, as well as the mutually exclusive conditions specified by the two choices. Thus, the model's conclusion reflects a logical error arising from semantic confusion.

### F.2   Commonsense Shortcut

```
User:

Context:
...
6. If Alvaro faces adversity, then he either stays motivated or pursues his goals, but not
    ↪ both.
...
Assistant:
...
Since Alvaro is facing adversity and is either staying motivated or pursuing his goals, it is
    ↪ clear that Alvaro is overcoming challenges in some form. Therefore, the statement "
    ↪ Alvaro does not overcome challenges" is false.
...
```

In this example, after confirming that Alvaro is facing adversity and satisfies either "staying motivated" or "pursuing goals", the model proceeds to conclude that "Alvaro overcame the challenge". Although this inference may seem reasonable from a commonsense perspective, it lacks a rigorous logical basis. The context of the question does not provide information on whether Alvaro actually overcame the challenge. Therefore, the model's conclusion is a flawed logical generalization based on a commonsense shortcut, which fails to adhere to the principles of deductive reasoning.

## G   Details and Credibility Analysis of the GPT Soundness Metric

We use a simple prompt to ask the GPT-OSS-120B model to evaluate the correctness of the reasoning path and ask it to return a JSON; when the reasoning is correct, it only needs to return a bool value, but when it is incorrect, it needs to provide the location and reason for the error.

**System prompt**

```
System:

The following is a problem statement and a reasoning chain to solve it.

Problem:
...

Reasoning:
...

Task:
Please evaluate whether the reasoning process contains any irreparable logical errors.
- If so, return a JSON where "is_error" is True, "error_instance" contains the original
    ↪ text of the first instance of the error, "error_type" specifies the type of the
    ↪ error, and "error_reason" explains the reason for the error.
- Otherwise, return a JSON where "is_error" is False.
Wrap your JSON within ```json ```.
```

We manually checked 10 outputs each from `Outcome-CoT` and *PRoSFI* to evaluate the correctness of GPT's assessment of the reasoning process.

For the outputs of `Outcome-CoT`, GPT's evaluations were all correct. Among them, 8 outputs had errors of invalid deduction, one output was completely correct, and one output had an error of misinterpretation of premise.

For the outputs of *PRoSFI*, GPT's evaluations were almost entirely correct but slightly too strict. Among them, 5 outputs had errors of invalid deduction, while 3 outputs were completely correct. The remaining two outputs had reasoning that was generally correct but contained some minor errors: the first output used unnecessary premises in one step, and the second output wrote the wrong name for the reasoning rule used.

## H    REWARD FUNCTION FOR *PRoSFI* STEP-BY-STEP VERIFICATION

We first extract a structured JSON representation from the natural language reasoning paths generated by the model. We then determine whether the format is correct based on its validity and the structural specifications defined in the main text. By analyzing the `id` and `conclusion` fields of the last object in the JSON, we can determine whether the final answer is correct.

To verify the logical correctness of the entire reasoning process, we use a top-down syntax parser to parse the first-order logic (FOL) expressions generated by the model. These are then converted into formats supported by target formal tools such as Lean4, Prover9, or Z3. For each reasoning conclusion, we express it as a proposition in Lean4 and verify it using a sequence of tactics or a large language model capable of formal theorem proving. Alternatively, the reasoning steps can be converted into input formats accepted by Prover9 or Z3, and the validity of each step can be verified using an automated solver.

The algorithm is presented in Algorithm 1, which follows this routine:

1. Extract a summary and FOL statements from the model's response.
2. If no valid statements are extracted, return a score of 0.
3. If the ID of the final reasoning step doesn't match the ground truth, return a score reflecting formatting issues.
4. If all FOL problems are solved correctly, return the full score.
5. Otherwise, return a partial (optional) score.

## I    CORRELATION ANALYSIS OF THE RESULTS ON `OUTCOME-COT`

For the output of the `Outcome-CoT` method, we analyze the relationship between two available metrics: the correctness of the final answer and the reasoning soundness automatically evaluated by GPT (GPT Soundness). The confusion matrix is shown in Figure 4.

---

**Algorithm 1** Reward Function for *PRoSFI* Step-by-Step Verification

---

**Require:** response, ground_truth
**Output:** score
 1: summary ← `extract_summary(response)`
 2: statements ← `extract_fol_problems(summary, ground_truth)`
 3: **if** statements is None **then**
 4:     return 0.0
 5: **end if**
 6: **if** summary.last_id $\neq$ ground_truth **then**
 7:     return `format_score`
 8: **end if**
 9: **if** `fol_all_solved(statements)` **then**
10:     return `full_score`
11: **else**
12:     return `option_score`
13: **end if**

---

It can be observed that a large number of reasoning paths, although yielding the correct answer, are evaluated as unsound in their reasoning process. This result indicates that a correct answer alone is not sufficient to reflect the logical validity of the reasoning process. It further supports the qualitative observation in Section 5.2 — namely, that traditional CoT methods struggle to ensure the validity of reasoning paths.

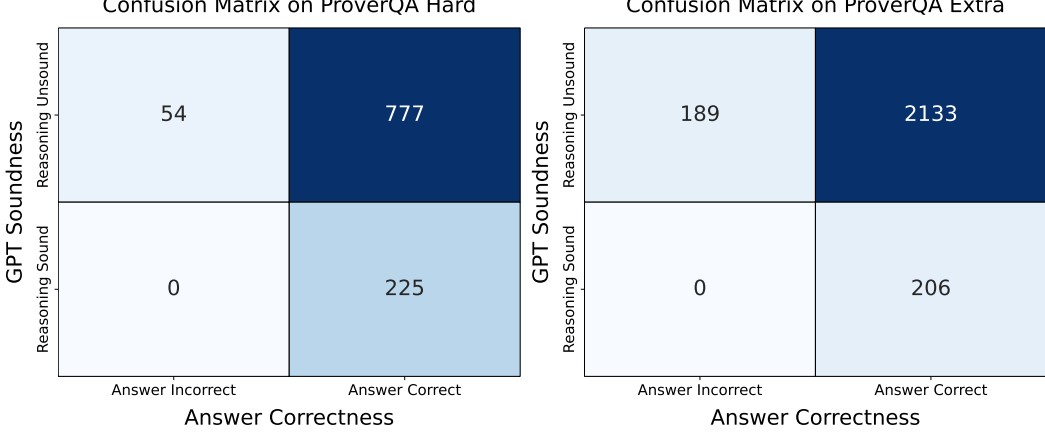

Figure 4: **Left:** Confusion matrix of answer correctness vs. reasoning soundness for `Outcome-CoT` on the `ProverQA-hard` subset. **Right:** Corresponding results on the `ProverQA-extra` subset. The lack of alignment between correct answers and sound reasoning indicates that answer accuracy alone is not a sufficient criterion for judging the quality of reasoning.

In comparison, Figure 3 presents the heatmaps of variable correlations on the two subsets. The results show that the designed reward signal has a stronger correlation with GPT Soundness than with the final answer correctness. This suggests that our formal verification mechanism effectively identifies and enhances logical rigor, making it a more reliable signal for training and selection.

## J    FAILURE CASE ANALYSIS

Here we analyze an erroneous instance in ProverQA dataset and its impact.

In the instance, two key propositions in the reasoning process are: "Hope lives in the wild." and "If Hope is in the wild, then she is either caught or released, but not both." However, the former is formalized as lives_in_wild(Hope) while the latter is formalized as in_wild(Hope) → (is_caught(Hope) ⊕ is_released(Hope)).

In the responses generated by the ProSFI model, a significant portion correctly identified that these two predicates represent the same proposition and arrived at the correct answer. However, this predicate mismatch resulted in no responses that could pass formal verification, leading to incorrect lower rewards for the responses. For example:

```
...
From 'h26: in_wild(Hope) → (is_caught(Hope) ⊕ is_released(Hope))', and using 'h2:
    ↪ lives_in_wild(Hope)', we derive 's9: is_caught(Hope) ⊕ is_released(Hope)'.
...
{
    "id": "s9",
    "dependencies": ["h26", "h2"],
    "conclusion": "is_caught(Hope) ⊕ is_released(Hope)",
    "rule": "Modus Ponens"
}
...
```

There were also some responses that failed to recognize the equivalence between the two predicates and mistakenly treated other similar predicates as equivalent, ultimately arriving at incorrect answers. For example:

```
...
From 'h20: captured_by_scientist(Hope) → (lives_in_lake(Hope) ∧ studied_by_scientist(Hope))',
    ↪  and using 'h6: is_caught(Hope)', we know 'lives_in_lake(Hope)'.
...
```