# OpenReview forum: "LEARNING TO GENERATE FORMALLY VERIFIABLE STEP-BY-STEP LOGIC REASONING VIA STRUCTURED FORMAL INTERMEDIARIES"
_ICLR.cc/2026/Conference — Submitted to ICLR 2026_

### Official Review · Reviewer_MNu4 · 2025-11-01

**Soundness:** 3
**Presentation:** 3
**Contribution:** 2
**Rating:** 6
**Confidence:** 4

**Summary:**

Authors identify that purely outcome-trained LLM reasoning may be correct but logically unsound.

Authors identify challenges with existing approaches to augmenting LLMs with formal verification
1. models excelling at NL reasoning struggle with formal code generation
2. models excelling at formal code generation struggle with NL inputs

They devise a new framework that prompts the model to generate structured logical information for each reasoning step that can be converted to formal propositions and verified externally. This reduces complexity on the LM generation side and proof construction side (both deal with decomposed problems).

From here, the authors use verification score to provide more granular reward to the model in RL and demonstrate improvements in reasoning soundness on logic tasks and knights and knaves puzzles.

**Strengths:**

- thorough baseline comparisons with direct proof generation and a RL baseline (outcome only)
- method is well motivated
- solid results in improving reasoning soundness
- compatible with test time scaling with DTV
- proposed reward correlates strongly with soundness

**Weaknesses:**

- negative impact to accuracy and OOD generalization in accuracy: exchanged for substantial boosts to reasoning soundness, but in OOD settings, the correlation between soundness and accuracy is low, so unclear if the trade off is valuable in this situation where you hope to use soundness as some proxy (per DTV).
- does not evaluate against less symbolic tasks (presumably because evaluation is more difficult, but it would be very interesting to see if this training benefits more realistic tasks)

**Questions:**

- Is there an interpretation/hypothesis for why PRosFI's accuracy drop from in-domain to out of domain is steeper than the outcome method? I had expected sound reasoning to be more transferable in new settings compared to the potentially incorrect shortcuts learned in outcome-based RL.
- Was there experimentation with penalizing the model more for the correct answer/incorrect reasoning case? It seems some positive reward is still rendered. If the goal is to optimize correctness, does this run slightly counter? Or is this for stability reasons?

---

> ### Author Response · Authors · 2025-11-24
> **Thank you for the supportive review and helpful questions. Concerns are carefully addressed.**
>
> Dear Reviewer, Thank you for the supportive review and helpful questions. We respond to the concerns point-by-point.
>
> ## On the accuracy or OOD generalization
>
> The reported OOD accuracy for PRoSFI is slightly lower than the outcome baseline in some settings. We investigated the slightly lower OOD accuracy on Knights-and-Knaves (8 people) and found two non-fundamental causes:
>
> 1. **Evaluation protocol**: our accuracy metric excluded samples with formatting errors. Some of these had minor format issues (e.g., overly many dependencies) but would be correct under a more permissive parser; additionally, a few long responses were truncated. This biases acc downward.
> 2. **Training detail**: DAPO’s overlong penalty speeds up convergence but tends to shorten responses, reducing available reasoning steps in harder OOD cases. Replacing it with simple overlong filtering increased average response length and improved OOD accuracy.
>
> After fixing both issues, the accuracy gap between ProSFI and the outcome-reward baseline becomes negligible. This suggests the observed tradeoff is not intrinsic to ProSFI but due to evaluation/training specifics. We will add this analysis and updated results.
>
>
>
> ## Lack of evaluation on less-symbolic tasks
>
> This is an important direction. PRoSFI relies on having a checker for intermediate steps, so we focused on domains where such verifiers are available and reliable, enabling clean causal conclusions about soundness. Extending to less-symbolic or open-world reasoning likely requires richer intermediate schemas tailored to the domain. We will explicitly state this limitation; In this sense, extending to formal/mathematical reasoning could be a natural next target.
>
> ## Why might the ID→OOD accuracy drop look steeper for PRoSFI?
>
> Beyond the two concrete issues above (see the response to the OOD generalization), a plausible *mechanistic* explanation is that under distribution shift (more entities / longer logical chains), the model incurs more truncated long responses, which hurt acc under a strict parser. Importantly, when intermediate steps *are* verified, their soundness transfers well; and using verification as a test-time selector restores strong scaling, as shown in Figure 2. We will add this interpretation in the revision.
>
> ## Reward for “correct answer but some steps unverified”
>
> We did experiment with removing this reward, i.e., assigning near-zero reward when the final answer is correct but some steps fail verification. In that setting, the reward hit rate changed little, but accuracy dropped noticeably. We therefore keep a small positive reward for this case to:
>
> * stabilize training early on (when verification errors are common), and
> * maintain pressure toward final correctness while still strongly favoring fully verified chains.
>
> This is standard reward-shaping: the hierarchy still penalizes unsound reasoning heavily relative to verified reasoning, but avoids collapsing learning when verified paths are rare.
>
> ----
>
> We’ll incorporate these clarifications and updates in the final version.

---

### Official Review · Reviewer_ZdD7 · 2025-11-03

**Soundness:** 2
**Presentation:** 2
**Contribution:** 1
**Rating:** 2
**Confidence:** 3

**Summary:**

This paper proposes PRoSFI (Process Reward over Structured Formal Intermediates), which is a method for fine-tuning LLMs on logical reasoning tasks. The core idea is to train the LLM to output intermediate reasoning steps in a structured formal language in addition to the final answer (another important observation is that outputting intermediate steps in this manner is easier than generating full formal proofs directly, which almost never succeeds for weaker LMs). These formal intermediate steps are then checked by a deterministic prover. The model is trained with GRPO, where it receives positive "stepwise formal rewards" for each intermediate step that is formally verified.

**Strengths:**

Flawed or unsound reasoning in LLMs is still an critical issue. The goal of improving the verifiability of intermediate reasoning is important. Towards this end, the approach taken by the paper is reasonable: using structured formal intermediate steps as a bridge (both to improve step by step verification and help weaker LLMs that are unable to produce full proofs that actually align with the CoT reasoning) is a logical and sound design choice for the considered tasks.

**Weaknesses:**

Using "stepwise formal rewards" is presented as a new paradigm, but it is not: this appears to be a clear instantiation of a process reward model, which is well-established in the alignment literature. Here the process reward is simply the binary signal from the deterministic formal verifier run on each reasoning step. While this is a reasonable reward and approach, the paper is not properly positioned relative to this existing literature.

I also find the empirical results to be a bit unconvincing. They consistently show that PRoSFI is less accurate than the Outcome-CoT baseline. While the "GPT Soundness" metric does increase, the merit of this trade-off is unclear and not fully justified.

**Questions:**

The "reward hit rate" of the direct Lean generation is really poor, which is a primary justification for your intermediate approach. This seems true for smaller models, but not so for larger ones. Do you expect the benefits of PRoSFI to scale?

Regarding the Test-Time Scaling (DTV) Comparison: In Section 5.5, you demonstrate that PRoSFI scales well with test-time verification (DTV) while Outcome-CoT does not. You attribute this to Outcome-CoT "lack[ing] the structured outputs required" for formal verification. This seems to preclude the possibility of using other types of verifiers. Why was a different verification method not considered for the Outcome-CoT baseline, such as a separate, learned verifier trained on its natural language reasoning steps (as is common in process supervision literature)?

Re generalizatio: the paper claim "strong logical consistency and reliable generalization" on the Knights and Knaves dataset. However, Table 3 shows that PRoSFI is less accurate than the Outcome-CoT baseline on the primary OOD split (85.31% vs. 89.75%). This mirrors the results from ProverQA. Can you comment on what leads to this drop in performance despite the more reliable generation?

---

> ### Author Response · Authors · 2025-11-24
> **Thank you for the detailed feedback. Each concern has been carefully addressed.**
>
> Thank you for the detailed feedback. We respond to each concern below.
>
> ## Novelty / positioning vs. process reward literature
>
> Indeed, “process supervision” and process reward models are established ideas. Our contribution is *not* to claim process rewards are new in general, but to introduce a **new way to instantiate process rewards using a structured formal intermediate language**:
>
> * Prior uses of formal languages in LLM reasoning mainly fall into two categories:
>   (i) using symbolic engines to *solve* problems, or
>   (ii) using formal verifiers to *check final answers or proofs*.
> * To our knowledge, **we are the first to use formal verification signals as *stepwise RL rewards*** by requiring each intermediate natural-language step to be accompanied by a machine-checkable structured dict, then rewarding only verified steps.
>
> We will revise the paper to clearly position PRoSFI as a *formal-intermediate instantiation of process rewards*, and cite/contrast with process supervision / PRM literature.
>
> ## Empirical results are a bit unconvincing, Accuracy vs. soundness trade-off
>
> We do not view the results as an inherent trade-off. After submission we analyzed why PRoSFI sometimes appears slightly lower in acc:
>
> 1. **Evaluation protocol**: our accuracy metric for ProSFI excluded samples with formatting errors. Some of these had minor format issues (e.g., overly many dependencies) but would be correct under a more permissive parser; additionally, a few long responses were truncated. This biases acc downward.
>
> 2. **Training detail**: DAPO’s overlong penalty speeds up convergence but tends to shorten responses, reducing available reasoning steps in harder OOD cases. Replacing it with simple overlong filtering increased average response length and improved OOD accuracy.
>
> After fixing these issues, the accuracy gap between ProSFI and the outcome-reward baseline becomes negligible. This suggests the observed tradeoff is not intrinsic to ProSFI but due to evaluation/training specifics.  We will add corrected evaluation and updated numbers in next revision.
>
> ## Do PRoSFI benefits scale to larger models?
>
> We expect the benefits to persist because the intermediate approach addresses **limitations that model scale alone does not fully remove** for direct Lean supervision:
>
> * **Cold-start dependence**: direct Lean RL requires substantial aligned formal data.
> * **Much larger model**: direct lean RL only works for much larger models (>72B) after large volumes of cold-start formal data, as shown in Kimi Prover and DeepSeek Prover V2.
> * **Alignment difficulty**: aligning those proofs to *natural-language intermediate steps* remains hard.
>
> In contrast, PRoSFI works with small models (7B) and no cold-start formal proofs, showing the supervision signal is accessible earlier in scaling. Scaling helps, and PRoSFI targets structural/alignment bottlenecks beyond raw capacity.
>
> ## Why not use an NL verifier for Outcome-CoT in DTV?
>
> Training a learned verifier over natural-language steps is a valid parallel direction, but it is **not yet a reliable drop-in baseline** here:
> NL step verifiers for *soundness/completeness* (not just final correctness) are still imperfect and prone to reward hacking or distribution shift.
>
> More importantly,  the ceiling of  Outcome-CoT (Best-of-N (BoN))  on soundness is far weaker than PRoSFI’s verification-based selection. Using our formal reward as a test-time selector yields strong scaling that BoN of Outcome-CoT cannot reproduce.
>
> Concretely, on ProverQA we observe:
>
> | n                             |     1 |     2 |     4 |     8 |    16 |
> | ----------------------------- | ----: | ----: | ----: | ----: | ----: |
> | Outcome-CoT BoN Soundness     | 21.31 | 26.56 | 32.48 | 37.94 | 40.91 |
> | PRoSFI (max-reward selection) | 74.05 | 77.75 | 81.75 | 85.00 | 86.35 |
> | PRoSFI BoN Soundness          | 74.05 | 81.94 | 88.60 | 92.91 | 93.94 |
>
> These results suggest the key driving force is *verifiability of intermediate steps*, not just sampling. We will clarify in the paper that NL-verifier baselines are interesting future work, but are not yet comparable in reliability to formal checking.
>
> ## Generalization claim vs. OOD acc drop
> The drop is explained by the same factors above (*Accuracy vs. soundness trade-off*), not by a failure of the method’s generalization mechanism.
>
> ---
>
> We appreciate the reviewer’s concerns on positioning and baselines; we will strengthen related-work framing, clarify the trade-offs/generalization issue, and add updated evaluations in the final version.

---

### Official Review · Reviewer_5EUo · 2025-11-03

**Soundness:** 3
**Presentation:** 4
**Contribution:** 3
**Rating:** 6
**Confidence:** 3

**Summary:**

The goal of this paper is to make the intermediate reasoning more reliable in long CoT’s, by using process supervision. Each intermediate step (in the natural language solution) is verified by a formal prover. However, writing formal proofs is not easy even for models that are good in natural language - the paper finds that it is very difficult to generate Lean formalizations that are aligned with the natural language intermediate steps. The ProSFI algorithm, introduced in this paper, introduces another reward function which is based on verification of the intermediate steps. For each intermediate step output by the LM, the model also outputs a JSON/YAML dictionary, which can be converted into a problem that a first-order logic verifier can solve.

The main experimental setting is as follows. They train and evaluate on ProverQA - a baseline is Qwen2.5-7B-Instruct trained only with an outcome/format reward, similar to Deepseek R1. They evaluate the “soundness” of the responses, i.e. how correct the intermediate responses are, using GPT-OSS-120B, and find that it is low for the outcome reward baseline. On the other hand, ProSFI improves the soundness significantly from the outcome-reward baseline, while preserving the correctness. Additionally, the ProSFI reward is highly correlated with the soundness as evaluated by GPT-OSS-120B (see Figure 3).

The other setting is the Knights and Knaves dataset, previously used in Logic-RL. In this work, both the outcome reward baseline and ProSFI are trained on problems with 3-7 people and tested on problems with 2-8 people. The soundness increases with ProSFI compared to the outcome reward baseline, while the accuracy is mostly similar.

**Strengths:**

1. The method for using process supervision is very interesting - they use JSON dictionaries to get around the limitations of small LMs when generating formal proofs.
2. The main results on ProverQA seem good.

**Weaknesses:**

1. The main results of this paper are for synthetic first-order logic tasks. There is a chance that there are additional challenges when trying to generalize to more challenging mathematical tasks.
    1. For example, the structure of the intermediate representations seems to rely on the nature of the task (which is first-order logic), because the intermediate dict specifies the logical rule.
    2. How can this be generalized to other mathematical settings? Will it still be easy to break up into atomic steps?
2. There is still a possibility that there is a tradeoff between rewarding for correctness and rewarding for soundness. This does not seem to be the case in the ProverQA setting, but in the OOD testing setting for Knights and Knaves, with 8 people, the accuracy is worse for ProSFI compared to the outcome reward method.

**Questions:**

1. Why do you believe the soundness is lower on ProverQA Extra, even with ProSFI?
2. Can it ever be the case that some steps are easy to reason about in natural language, but difficult to prove/verify formally? Have you encountered this?

---

> ### Author Response · Authors · 2025-11-24
> **Thank you for the thoughtful review and constructive questions**
>
> Thank you for the thoughtful review and constructive questions. We respond point-by-point below.
>
> ## Generalization beyond synthetic first-order logic
>
> The core idea of ProSFI—decomposing long reasoning into verifiable subgoals and applying process supervision—matches common Test-Time Scaling practice in math, where structured subgoal decomposition plus intermediate verification has shown strong gains such as in DeltaProver (https://arxiv.org/abs/2507.15225). We therefore believe the approach is feasible in wider math settings, though it may require more sophisticated cold-start strategies, training recipes, reward shaping, and engineering. We will clarify this scope and outline these future directions in the revision.
>
>
> ## On the potential correctness–soundness tradeoff
>
> We investigated the slightly lower OOD accuracy on Knights-and-Knaves (8 people) and found two non-fundamental causes:
>
> 1. **Evaluation protocol**: our accuracy metric for ProSFI excluded samples with formatting errors. Some of these had minor format issues (e.g., overly many dependencies) but would be correct under a more permissive parser; additionally, a few long responses were truncated. This biases acc downward.
> 2. **Training detail**: DAPO’s overlong penalty speeds up convergence but tends to shorten responses, reducing available reasoning steps in harder OOD cases. Replacing it with simple overlong filtering increased average response length and improved OOD accuracy.
>
> After fixing these issues, the accuracy gap between ProSFI and the outcome-reward baseline becomes negligible. This suggests the observed tradeoff is not intrinsic to ProSFI but due to evaluation/training specifics. We will add this analysis and updated results.
>
> ## Why soundness is still low on ProverQA Extra even with ProSFI
>
> The Extra split is generated with intentionally difficult settings: each problem may contain 20–30+ propositions, making full reasoning within the response length highly challenging. This lowers compile rate and thus overall soundness. Importantly, among chains that *do* pass ProSFI verification, soundness remains high, and ProSFI works well as a test-time filter, showing clear Test-Time Scaling behavior, as shown in Figure 2 (right). We will emphasize this in the revision.
>
> ### Natural-language-easy but formally-hard steps
>
> Yes, this can happen. Some steps feel “obvious” in natural language because they rely on implicit assumptions, semantic shortcuts, or commonsense gaps, but formal proof/verification requires those premises and rules to be made explicit, which is harder for models. We observed such cases, and they motivate our use of intermediate dictionaries plus FOL verification to bridge the alignment gap. We will add discussion and examples.

---

### Author Response · Authors · 2025-12-03
**Final response with an updated version of the paper**

We thank the reviewers and area chair for the comments and evaluation. Here is our final response with an updated version of the paper.

### **A common question raised by reviewers: The answer correct rate of ProSFI drops on the extra split in Table 2. Is there any accuracy vs. soundness trade-off?**

This phenomenon also puzzled us initially, but we do **not** believe there is an inherent trade-off between accuracy and soundness.  After carefully inspecting failure cases, we identified a key factor: **some data in ProverQA have quality issues**. A typical problem is that identical propositions in natural language are formalized using different predicates (see our new *Failure Case Analysis* section in the Appendix), a point that is also mentioned by the authors of ProverQA in their new paper: [https://arxiv.org/pdf/2506.04575](https://arxiv.org/pdf/2506.04575).

Motivated by this finding, we manually inspected and repaired a portion of the instances in the extra split (test set). On this repaired subset, the accuracy of the originally trained ProSFI and Outcome-CoT models is 90.80% and 93.00%, respectively, and the GPT Soundness rates are 40.10% and 11.40%, respectively. For comparison, in Table 2 on the original data, ProSFI reports 85.36% accuracy and 28.72% GPT Soundness. Thus, solely fixing the test examples yields absolute improvements of 5.4 and 11.4 percentage points in accuracy and soundness, respectively, for ProSFI. We note that Outcome-CoT is much less affected by these problematic formalizations, because it relies more heavily on the natural language statements than on the formal ones.

**This finding demonstrates that ProSFI can indeed substantially improve soundness while maintaining the accuracy rate.** A more principled comparison would require retraining and reevaluating all methods on a fully repaired version of ProverQA, which we expect would considerably if not fully reduce the accuracy gap between ProSFI and Outcome-CoT. We have therefore added a footnote to document this dataset issue and are working to complete the full retraining and reevaluation within one more week.

### **Other revision in the updated version of the paper.**

- We emphasize that our main contribution is instantiating reinforcement learning with step-by-step rewards guided by formal provers, a direction that, to our knowledge, has not been addressed in the existing literature.

- We add a discussion of future work on extending our framework to mathematical reasoning and other application domains.

- We expand the related work section on PRMs to better position and clarify the contribution of this paper.

---

### Meta-Review · Area_Chair_wjWY · 2026-01-07

**Summary:**

This paper proposes PRoSFI (Process Reward over Structured Formal Intermediates), which is a method for fine-tuning LLMs on logical reasoning tasks. The core idea is to train the LLM to output intermediate reasoning steps in a structured formal language in addition to the final answer (another important observation is that outputting intermediate steps in this manner is easier than generating full formal proofs directly, which almost never succeeds for weaker LMs). These formal intermediate steps are then checked by a deterministic prover.

The main concerns of reviewers, which I agree with include the lack of a natural language PRM baseline or scaling up their approach to broader NL tasks, the decrease in accuracy as we use the ProSFI method, and results with bigger models or harder benchmarks. The authors did say they were working on retraining the approach, but did not get back on the results finally before the rebuttal. The details of how the test set was repaired were also not clear. I believe that the significance of this paper would not be clear unless these comparisons are established, and we know why ProSFI drops clearly + proper experiments are performed to see if it can generally match outcome-CoT on both ID/OOD accuracy.

I also feel that comparisons of other reward designs, and more analysis (e.g., which component of reward works when?) will be critical here as well. In particular, while the idea of using formal verification is definitely useful for soundness, it would be good to make these comparisons and analyses to make the paper solid.

**Reviewer Concerns:**

In general, Reviewer ZdD7's concerns were not fully addressed, especially in regards to why GPT-soundness makes sense as a metric, NL process verifiers comparison, and accuracy vs soundness tradeoff. These are also the concerns I have with the rebuttal and the paper right now.

For the other reviewers, I believe the concerns may have been addressed but the above points do remain.

**Reviewer Scores:**

Reviewer ZdD7 might have moved upto a 4. The rest of the scores might remain the same.

---

### Decision · Program_Chairs · 2026-01-26

Reject